

# Quantifying the impacts of wildfires on soil thermal, hydrological and carbon dynamics in northern Eurasia from 2003 to 2016

Yiming Xu[1], Qianlai Zhuang[1], Bailu Zhao[1], Michael Billmire[2], Christopher Cook[2], Jeremy Graham[2], Nancy HF French[2], Ronald Prinn[3]

[1] Department of Earth, Atmospheric and Planetary Sciences, Purdue University, West Lafayette, IN 47907 USA
[2] Michigan Tech Research Institute, Michigan Technological University, Ann Arbor, MI 48105 USA
[3] Department of Earth, Atmospheric, and Planetary Sciences, MIT, Cambridge MA

*Correspondence to*: Qianlai Zhuang (qzhuang@purdue.edu)

To be submitted to Biogeoscience

**Abstract.** We use a process-based biogeochemistry model to simulate the fire impacts on soil thermal and hydrological dynamics and carbon budget of forest ecosystems in Northern Eurasia during 2003-2016 based on satellite-derived burn severity data. We find that burn severity generally increases in this region during the study period. Simulations indicate that fires increase soil temperature by 0.2-0.5 °C through removing the ground moss and surface soil organic matter, especially in Asian part of the region. Fires also increase water runoff by about 131 million $m^3$ $yr^{-1}$ through reducing post-fire evapotranspiration, leading to a higher regional river discharge. Fires remove 1.7 Pg C of ecosystem carbon through combustion emissions during this period and reduce net ecosystem production from 106.4 to 66.1 Tg C $yr^{-1}$. Fires lead the forest ecosystems to lose 2.3 Pg C, shifting the forests from a carbon sink to a source in this period. Our study highlights the importance of wildfires in affecting soil thermal and hydrological and carbon dynamics in boreal forests.

## 1 Introduction

Boreal forests store more than one-third of the global terrestrial carbon, playing an important role in the global carbon cycling (Kasischke & Stocks., 2000). This large carbon pool is vulnerable to wildfire disturbance (Helbig et al., 2016). Fire can release large amounts of carbon into the atmosphere through direct combustion emission and decrease post-fire ecosystem production (Kurz & Apps, 1999; Amiro et al., 2006; Yin et al., 2020; Zhao et al., 2021), thereby increasing carbon dioxide concentration in the atmosphere. Recent studies show that burn area and burn severity are increasing in boreal forests under the anthropogenic climate warming (Gillett et al., 2004; Zheng et al., 2021; Iglesias et al., 2022), suggesting there are positive feedbacks between fires and the global climate (Moubarak et al., 2023).

Previous studies show that the river discharge in Eurasia high latitude regions has been increasing (Peterson et al., 2002; McClelland et al., 2006; Overeem & Syvitski, 2016). This change might be due to permafrost thaw (Lamontagne-Hallé et al., 2018; Wang et al., 2021) and wildfires. Wildfires can influence soil thermal and hydrological regime. Specifically, wildfires change the soil hydrology of forests by removing vegetation canopy,



reducing evapotranspiration (Poon & Kinoshita, 2018). They can also enhance post-fire water runoff (Moody & Martin,
2001; Thomas et al., 2021) by changing vegetation and soil texture (Shakesby & Doerr, 2006; Moody et al., 2009;
Ebel & Martin., 2012). These dynamics are also affected by varying burn severities (Moody et al., 2008). Previous
studies show that fire can increase soil surface temperature through direct heat release (Franklin et al., 1997; Debano
et al., 1998, Certini, 2005) and influence the energy balance of soil surface (Li et al., 2017; Zhao et al., 2021) by
changing the albedo and evapotranspiration of the vegetation.
Wildfires impact the carbon cycle of boreal forests by removing aboveground vegetation and consuming soil organic
matter (Turquety et al., 2007; Turetsky et al. 2011, de Groot et al., 2013; Rogers et al., 2015). Soil organic matter
combustion can release large amounts of carbon into the atmosphere, accounting for up to 90% of the total carbon
emission for severe fires (Walker et al., 2018). In addition to direct combustion emissions, post-fire soil respiration
can also impact the carbon budget due to changing soil temperature and moisture (Kulmala et al., 2014). After fire,
there might also be a vegetation shift (Denslow, 1980; Gewehr et al., 2014; Stuenzi et al., 2022). With the change in
ecosystem structure, soil moisture and soil temperature, carbon dynamics can be significantly affected (Sullivan et al.,
2011; Li et al., 2017).
Previous studies have modeled fire's influence on soil physical properties and carbon dynamics with limitations. Many
of them mainly focus on direct combustion emissions (Conard & Ivanova, 1997; Amiro et al., 2001; French et al.,
2002). Some studies are on site level and lack regional quantification (Moody qt., 2008; Van Eck et al., 2016; Poon
& Kinoshita, 2018). Further, existing studies typically have not incorporated the effects of burn severity because its
data are unavailable or difficult to obtain (Kasischke et al., 2005; Balshi et al., 2007; Van Eck et al., 2016). Some
studies have tried to overcome these limitations by using process-based models (Zhao et al., 2021), but have not
focused on analyzing soil hydrological regime, especially in Eurasia high latitudes. These modeling studies could be
improved by using burn severity information to holistically understand how fires affect soil thermal and hydrological
and carbon dynamics.

In this study, we use a sophisticated process-based model, the Terrestrial Ecosystem Model (TEM) (Zhuang et al.,
2002; Zhao et al., 2020; Xu et al., 2024) to quantify the influence of fire on regional soil thermal, hydrological, and
carbon dynamics in Eurasia northern high latitude forests. With satellite-derived burn severity data, we analyze these
impacts from 2003 to 2016.

## 2 Data and method

### 2.1 Burn severity data

Fire burn area perimeters were obtained from the Global Fire Atlas (Andela et al, 2019), which were derived from
MODIS daily moderate-resolution (500 m) Collection 6 MCD64A1 burned-area data from 2003 (the start of the





MODIS mission) to 2016. To reduce the Global Fire Atlas down to the areas of interest, annual detected burn area
shapefiles were first input to the QGIS Repair Geometry tool to clean geometries, then clipped to the extents of Europe
and Asia north of 45 degrees, for each year of 2003-2016. Batch processing capabilities available within QGIS were
then used to reproject shapefiles to Asia/Europe Albers Equal Area Conic and remove all fire shapefiles of less than
2 hectares. Overlapping fire detections within a single year were combined using QGIS 'Dissolve' and allowed for
keeping of associated polygon information.
Because the Global Fire Atlas is based on MODIS fire detections, the available fire shapefiles had an inherent
blockiness and holes within fire perimeters. The geometry was simplified by reducing the number of vertices but
thresholding to not change any features by more than 250 meters, the native resolution of the burned area product.
Interior holes within the perimeters were filled in to further simplify the geometry. This allowed for more feasible
processing times for subsequent buffering operations. Many of these holes were water bodies where values would be
removed from the analysis by optical cloud masking. In the cases where the holes were not water, it was considered
that they could either be real unburned areas within the perimeter, or false negatives that actually burned but not hot
enough to trigger the active hotspot detection. However, it was determined that these scenarios would balance out at
the scale of the analysis.
A Google Earth Engine (GEE) script was developed to calculate a pre- and post- Normalized Burn Ratio (NBR) value
from within each FIRE perimeter, as well as from within a 300 m buffer ring offset 1.5 km from the perimeter. Data
were sourced from Landsat 4-8 collections and used median pixel values after applying a cloud, cloud shadow, snow,
and water mask. NBR for each fire was calculated as the ratio between the near infrared (NIR) and short-wave infrared
(SWIR) bands for the available Landsat mission:
$NBR = \dfrac{NIR - SWIR}{NIR + SWIR}$   (1)
Pre-fire data use the median pixel values of the image collection, with images filtered to the approximate snow-free
fire season of June 15 to September 15, from the two years before the fire. Post-fire data use the median pixel values
from the image collection June 15 to September 15 the year following the fire. The median pixel values from within
the perimeter and buffer ring averaged to create a pre-fire NBR and post-fire NBR for each perimeter and buffer ring.
These NBR values were written out of GEE along with each unique fire ID. Methods in the GEE code for masking
and pooling the Landsat 4-8 data were sourced from Holsinger et al. (2021).

The values in the buffer ring are used to correct differences between the imagery not related to the fire (e.g., phenology,
plant health), as this land area is assumed to have not burned. One caveat is that in some areas that had nearby fires
within the same year, there may be some overlap between the buffer ring and another fire perimeter, however these
cases are uncommon. The difference between the pre-fire NBR - the postfire NBR is known as the delta NBR (dNBR)
and can be used to estimate burn severity for a fire:
$dNBR = NBR_{prefire} - NBR_{postfire}$   (2)



In addition to calculating dNBR for each individual fire over 2003-2016, data were aggregated at a 0.5°×0.5° grid cell
for each year in order to incorporate these data into modeling.
**2.2 Model description**
We use the Terrestrial Ecosystem Model (TEM; Zhuang et al., 2002) that simulates soil thermal, hydrological regime
and carbon dynamics in terrestrial ecosystems. TEM has been used to simulate fire disturbance on carbon dynamics
and soil thermal regime in North America (Zhuang et al., 2002; Zhao et al., 2021; Xu et al., 2024). Apart from direct
carbon emission due to combustion, TEM can also simulate post-fire carbon dynamics during the ecosystem recovery
(Fig. 1; Zhao et al., 2021; Xu et al., 2024).

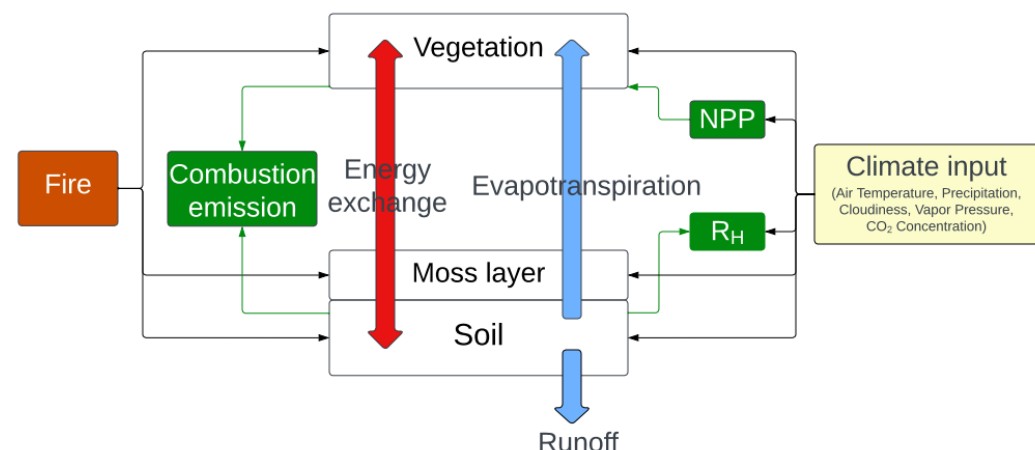

**Figure 1: Fire effects on forest ecosystem dynamics of energy, water, and carbon.**
The calculated delta Normalized Burn Ratio (dNBR) values that represent burn severity for fires are used to estimate
the proportion of vegetation and soil carbon consumption in TEM (Zhuang et al., 2002; Zhao et al., 2021). In the
model, fire removes carbon from vegetation and soil based on the burn severity. First, TEM estimates the composite
burn index (CBI) based on dNBR values:
$CBI = p_1 \times dNBR + p_2$   (3)
where $P_1$ and $P_2$ are parameters calibrated in the previous study (Zhao et al., 2021). With estimated CBI value, the
removed carbon is calculated based on:
$R = p_3 \times CBI + p_4$     (4)
where R is the removed ratio, $P_3$ and $P_4$ are parameters given by field measurement (Boby et al., 2010).



Fires with greater burn severity remove more carbon from the ecosystem. Following fire, the reduction in litter-fall
might further impact soil organic carbon composition during the vegetation recovery. The reduced vegetation carbon
storage can result in lower ecosystem production during the recovery phase.
Fire also impacts soil temperature and soil water content. In TEM, soil thermal regime is simulated with surface air
temperature and land surface energy exchange process that is influenced by the canopy and moss (Xu et al., 2024).
Simulated vegetation canopy and ground moss layer can impact soil physical properties. Fire removes ground moss
layer which serves as an insulation in the soil thermal module. In the year of fire, TEM assumes that the fire completely
combusts the moss. After fire, the moss thickness recovers as the following:
$moss = a_1 \times (1 - e^{b_1 \times t})$    (5)
where moss is the moss thickness (cm), t is the year after fire, $a_1$ and $b_1$ are parameters. Also, canopy can influence
the surface energy budget and soil surface temperature. The soil surface energy budget is estimated as:
$Q = SH + LH + SR + LR$    (6)
where Q is the energy budget, SH is the sensible heat flux, LH is the latent heat flux, SR is the short-wave radiation
and LR is the long-wave radiation. These variables are calculated based on the meteorological input data and leaf area
index. More details can be found in Xu & Zhuang (2023). The removal of ground vegetation by fire will change the
soil energy exchange and impact the soil temperature. On the other hand, soil temperature and soil moisture will affect
the growth of the vegetation. These influences on soil physical properties can further change carbon dynamics, such
as heterotrophic respiration ($R_H$), since these processes are highly correlated with soil thermal regime.
Soil water content and runoff are estimated with precipitation, evaporation and the soil texture. Fire-derived changes
of vegetation will affect the evaporation process and in turn impact the soil hydrological regime. In TEM, transpiration
is calculated as (Zhuang et al., 2002):
$$trans = \frac{slope \times drad + c_p \times pa \times \frac{vpd}{ra}}{\left(slope + gamma \times \left(1 + \frac{rc}{ra}\right)\right) \times le} \times dayl \quad (7)$$
where *trans* is transpiration (mm/month), slope is the slope of vapor pressure curve, *drad* is the canopy daily average
radiation, a function of the leaf area index. $C_p$ is the specific heat of air (J kg$^{-1}$K$^{-1}$), *pa* is the density of air (kg m$^{-3}$),
*vpd* is the vapor pressure deficient from canopy to air (mbar), *ra* is canopy aerodynamic resistance, *rc* is canopy
resistance to water vapor, *le* is latent heat of vaporization of water (J kg$^{-1}$), *dayl* is the length of a day (s) and *gamma*
is a parameter. The water exceeding the soil capacity is calculated as runoff. In the model, fire removes ground
vegetation, the amount of removal depends on the burn severity. The reduced leaf area index leads to lower
evapotranspiration and higher soil water content, resulting in higher runoff.
**2.3 Input data and simulation protocols**
Monthly mean surface air temperature, cloudiness, precipitation, vapor pressure, and surface wind speed from ERA5
(Hersbach et al., 2020) during 2003-2016 are used to drive the model. Spatially-explicit soil texture (percentage of
silt, clay and sand), elevation, plant function types and annual $CO_2$ concentrations of the atmosphere are also used
(Melillo et al., 1993; Zhuang et al., 2002).



The model is calibrated in previously published work (Zhuang et al., 2002; Zhao et al., 2021). It can well estimate the
soil and vegetation carbon in three Canadian boreal sites (Zhao et al., 2021). The parameters used in this study are
adopted from this previous work.
The model is spun-up for 120 years before 2003 with cyclic climate data from 1988 to 2003 to achieve an equilibrium
state. Then transient simulations from 2003 to 2016 are conducted for each grid cell at a spatial resolution of $0.5°×0.5°$
for the Eurasia high latitude forests.
We conduct two regional simulations, with and without fire disturbance considered, respectively. Fire polygons are
dissected into each unit with unique fire history and then intersected with each grid cell when considering fire impacts.
The output values for each grid cell are area-weighted mean of each fire polygon and no-burn area within the cell
(Zhao et al., 2021). To test the sensitivity of the four parameters in equations 1 and 2, we also conduct four additional
simulations with modifications of the parameters. In each simulation, we increase one parameter by 10% and compare
the results with original simulations.
**3 Results**
**3.1 Changes in fire regime**

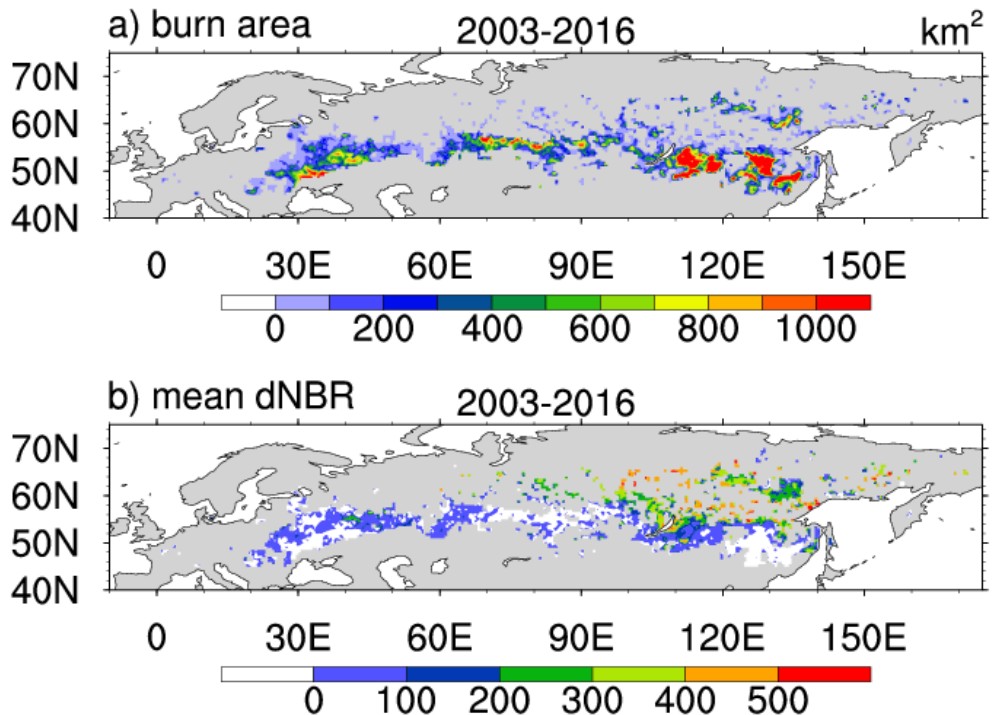






**Figure 2: Burn area (a) (units: km2) and severity (b) (delta Normalized Burn Ratio, dNBR values) in Eurasia forests during**
**2003-2016.**
Most grid cells are burned for less than 300 km$^2$ during 2003-2016 while the maximum area can be larger than 1000
km$^2$ for East Europe, Central Asia and Northeast Asia (Fig. 2). For the burn severity, the mean dNBR value is less
than 150 for most grid cells while it can be greater than 350 for most grids in East Russia. For those widely-burned
grid cells (Fig. 2a), the burn severities are generally low, with widely-spread dNBR values that are smaller than 0 (Fig.
2b). This means that high severity fires mostly occur in a small area, and mild fires take up more land during the study
period (Fig. 3a). There is a significant increasing trend of the regional mean burn severities from 2003 to 2016 (Fig.
3b) with a slope of 4.9 and a P value of 0.038.

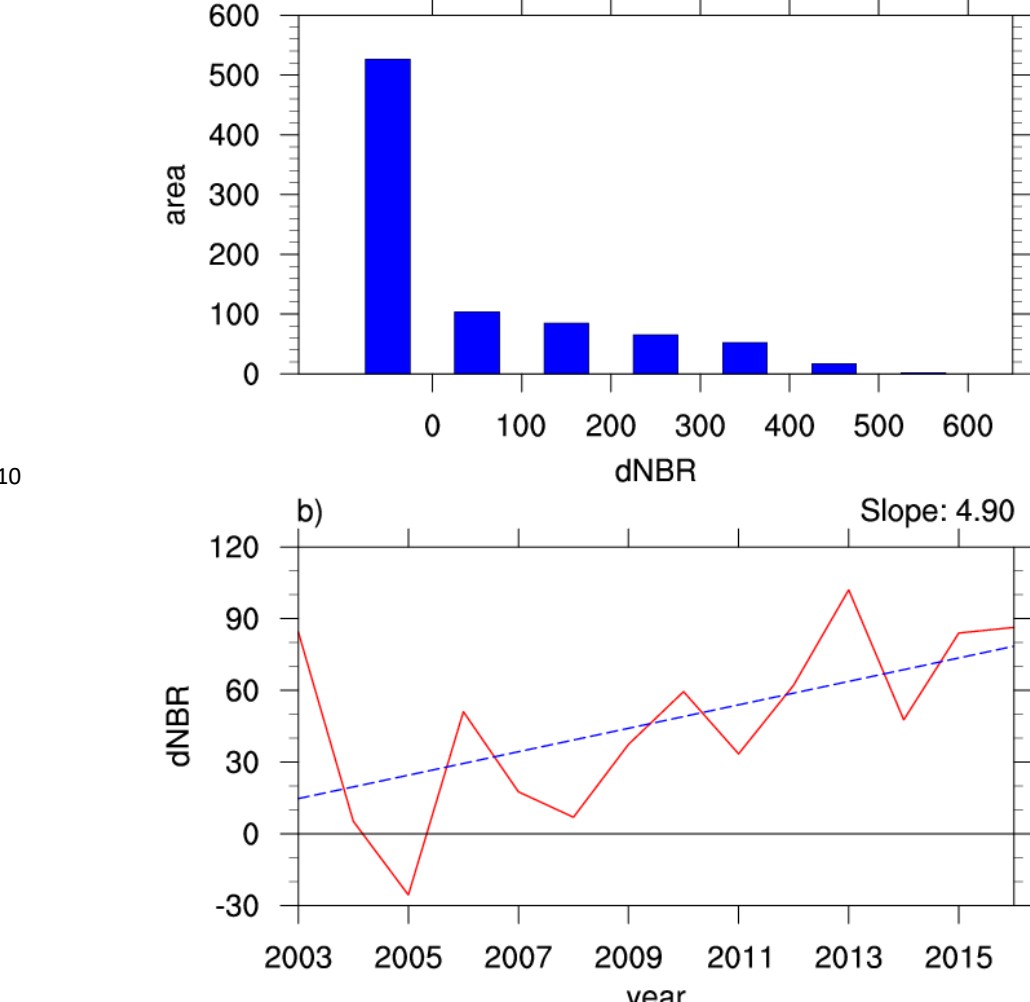




**Figure 3: Summary of fire regimes: (a) Histograms of dNBR. The heights of bars are total fire area in which average dNBR**
**is within the thresholds indicated by the x axis. (b) Annual area-weighted mean dNBR value (red line) and its linear trend**
**(blue line).**
**3.2 Fire impacts on soil thermal and hydrological regime**

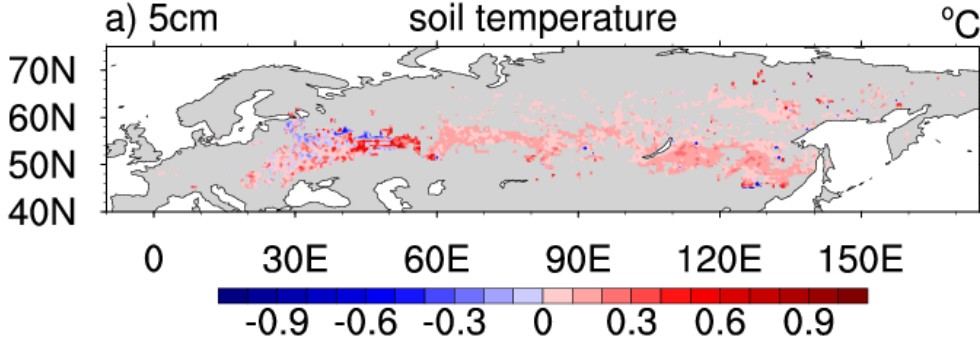

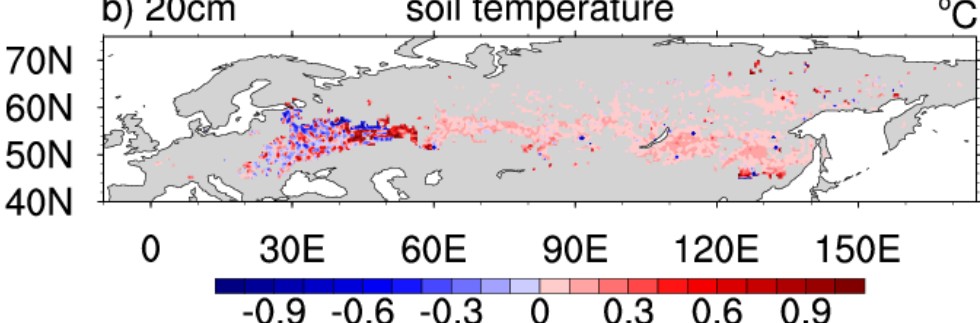

**Figure 4: Simulated mean soil temperature differences with fire minus without fire in Eurasia for 5 cm (a) and 20 cm (b)**
**depth during 2003-2016.**
Compared with no fire simulation, the model estimates higher mean soil temperature (Fig. 4) in Asia with a magnitude
of 0.2-0.5 $^{\circ}$C since fire can remove the ground moss layer which can insulate heat and cool the soil. The deep soil
layer temperature change follows the pattern of the surface soil but exhibits smaller magnitudes. There is no clear
pattern for soil temperature changes in Europe, especially for the 20 cm depth, since it is relatively warmer in Europe
and the model estimates little or no ground moss layer in this area.





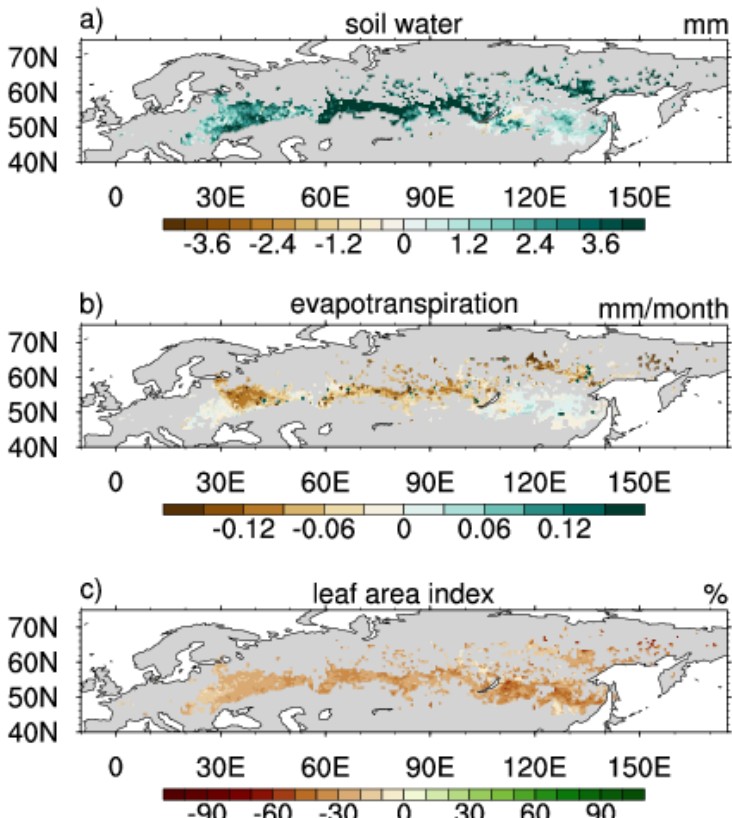

**Figure 5: Simulated mean soil water content (a, for top 70cm soil), evapotranspiration (b) and relative leaf area index (c)**
**differences due to fire (with fire minus without fire) during 2003-2016.**

Due to fire, the simulated mean soil water content increases for most grid cells with a magnitude of more than 2 mm
(Fig. 5a). The model also generally estimates lower evapotranspiration with a magnitude of 0.1 mm month-1 (Poon &
Kinoshita, 2018). These changes can be attributed to the leaf area index change (Fig. 5c). After fire, simulated leaf
area index decreases since fire significantly removes canopy and ground vegetation. Decreased leaf area index
weakens evapotranspiration, leading to higher soil water content. As a result, the model estimates higher water runoff
(Fig. 6) with a mean magnitude of around 131 million $m^3$ $yr^{-1}$ for the whole region. The changes in water runoff due
to fire show an increasing trend during 2003-2016, which is consistent with the annual mean burn severity trend (Fig.

11   3).





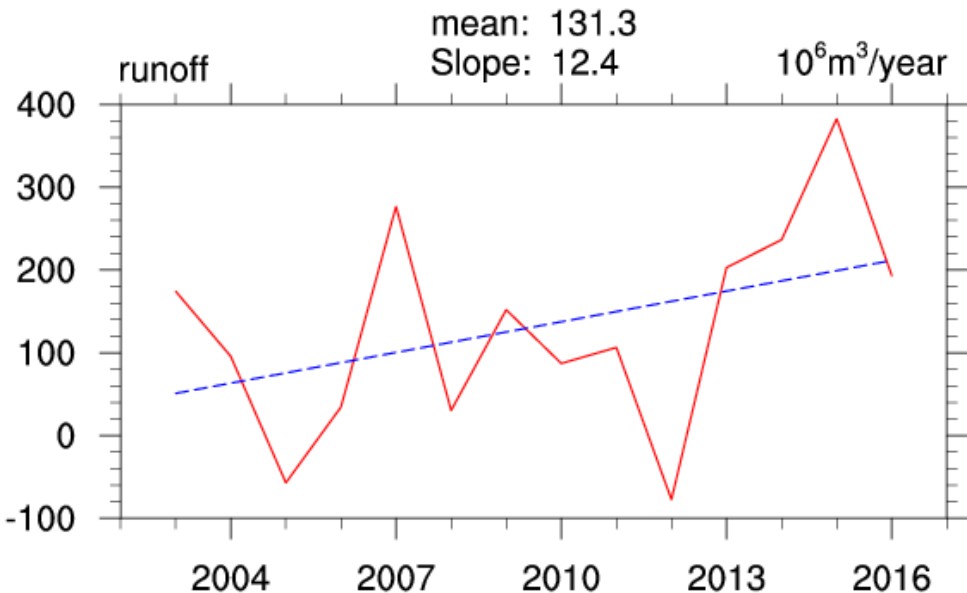

**Figure 6: Temporal evolution of regional mean runoff differences (with fire minus without fire) in Eurasia.**
**3.3 Fire impacts on carbon dynamics**
**Table 1: Regional mean NPP, RH, NEP and combustion emission in Eurasia during 2003-2016 for the two simulations.**

| Units: Tg C yr$^{-1}$ | Without fire | With fire |
|---|---|---|
| NPP | 1177.5 | 1132.0 |
| $R_H$ | 1071.0 | 1065.9 |
| NEP | 106.4 | 66.1 |
| Combustion | / | 121.9 |

When fire is not considered, the model estimates the whole region as a net carbon sink, with a mean net ecosystem
production (NEP) of 106.4 Tg C yr$^{-1}$ (Table 1). When fire is considered, the simulated net primary production (NPP)
is lower since fire significantly removes vegetation carbon and decreases the gross primary production (GPP). Despite
higher soil temperature due to fire, the simulated heterotrophic respiration ($R_H$) decreases since fire removes soil
carbon. The magnitude of the changes in NPP is greater than $R_H$, leading to a decreased NEP. In all, the modeled NEP
is 66.1 Tg C yr$^{-1}$ for the whole region, 40.3 Tg C yr$^{-1}$ less than no fire situation. Although the model estimates a positive
regional NEP, the simulated regional combustion emissions of 121.9 Tg C yr$^{-1}$ are much greater than NEP, resulting
in a net loss of carbon at 55.8 Tg C yr$^{-1}$ during 2003-2016.



## 4 Discussion

### 4.1 Fire and river discharge

There is an increasing trend for both annual mean burn severity and water runoff during 2003-2016 in Eurasia. This might have increased river discharge (Peterson et al., 2002; Rood et al., 2017; Ahmed et al., 2020; Feng et al., 2021) to the Arctic Ocean and decrease the ocean salinity (Peterson et al., 2002; McPhee et al., 2009). Our simulated runoff differences due to fire in major river watersheds in Russia indicate that all watersheds exhibit an increase in the mean runoff due to fires (Table 2). Yenisei, Ob and Lena, the three major rivers that drain into the Arctic Ocean, exhibit relatively larger discharges than other rivers. Increasing river discharge and decreasing salinity might further influence the ocean circulation, such as Atlantic Meridional Overturning Circulation (AMOC) (Liu et al., 2019), impacting the global climate.

**Table 2: Simulated mean runoff difference due to fire in 7 major river watersheds in Russia. (Units: million m3 yr-1).**

| Watersheds | Runoff Difference |
|---|---|
| Volga | 3.7 |
| Yenisei | 16.4 |
| Ob | 36.1 |
| Lena | 7.8 |
| Amur | 0.8 |
| Dnieper | 0.03 |
| Don | 12.6 |

### 4.2 Comparison with other studies for carbon dynamics

We estimate that the total combustion emissions are 121.9 Tg C yr$^{-1}$ for Eurasia high latitude forests during 2003-2016, which is slightly lower than that in previous studies focusing on Russia or Eurasia (Table 3). Our study only focuses on the forests in Eurasia high latitudes while these studies cover other vegetation types including tundra and grasslands for instance. Together with our previous estimates of fire emissions of 67.7 Tg C yr$^{-1}$ from North America (Xu et al., 2024), we estimate the total combustion emissions from northern high latitude forests are 189.6 Tg C yr$^{-1}$. This is close to, but a little lower than previous estimates (Table 3).

**Table 3: Combustion emission comparison with other studies (Units: Tg C yr-1).**

| Studies | Time | Region | Carbon emission |
|---|---|---|---|
| This study | 2003-2016 | Eurasia | 121.9 |
| Mouillot et al. (2006) | 1990-1999 | Eurasia | 166 |
| van der Werf et al. (2006) | 1997-2004 | Russia | 188.4 |





| Shvidenko et al. (2011) | 1998-2010 | Russia | 82.0 |
| Yue et al. (2016) | 1997-2009 | Pan-Arctic | 200 |
| Ponomarev et al. (2021) | 2002-2020 | Siberia | 80±20 |
| Zheng et al. (2023) | 2000-2020 | Eurasia | 149.8* |
| Zheng et al. (2023) | 2000-2020 | Pan-Arctic | 212.3* |

*Calculated based on the global combustion emission flux data.
**4.3 Sensitivity and uncertainty analysis**
TEM estimates the proportion of carbon removal by using dNBR values to estimate the CBI and corresponding carbon
removal. The relationship between dNBR values and CBI (Eqn. 1) and the relationship between CBI and the
proportion of carbon removal (Eqn. 2) are based on field study (Zhao et al., 2021; Boby et al., 2010). Thus, those
parameters in the equations might have uncertainties. By raising each parameter by 10%, the estimated regional
combustion emissions all exhibit a slight increase (Table 4). The rising of $P_3$ leads to a most significant increase of
the combustion emission by 7.4%. This parameter is the slope of the linear regression relationship between CBI and
the proportion of removed carbon due to fire. Considering that equation 2 has a relatively low explained variance ratio
for the regression analysis, especially for canopy carbon ($R^2$=0.15, Boby et al., 2010), the relationship between CBI
and the combustion proportion leads to the largest uncertainty in our study.
**Table 4: Combustion emissions of sensitivity test simulations and their relative change compared with the original**
**simulation. The simulation name indicates the modified parameter in EqnS. 1 and 2.**

| Simulation | Combustion emission (Tg C yr$^{-1}$) | Percentage change (%) |
| --- | --- | --- |
| $P_1$ | 126.6 | 3.9 |
| $P_2$ | 125.8 | 3.2 |
| $P_3$ | 131.0 | 7.4 |
| $P_4$ | 124.7 | 2.3 |

Using dNBR values to represent burn severity in the model can also cause some uncertainties since other factors such
as moisture, elevation and time of burn can also impact burn severity (Kasischke & Hoy, 2012; Tan et al., 2007; Zhao
et al., 2021). The relationship between dNBR and combustion proportion is established for black spruce dominated
forests and the model is originally calibrated with field data from Alaska. This might induce uncertainties when
modeling Eurasia high latitude forests since the vegetation type and climate condition in Eurasia can be different from
Alaska. Additionally, we have not modeled the vegetation shift impacts after fire (Gewehr et al., 2014; Stuenzi et al.,
2022). Thus, the vegetation recovery after fire is not explicitly modeled, which might affect the corresponding soil
thermal and hydrological dynamics, in turn, influencing carbon dynamics.



**4.4 Limitations to this study**

First, in modeling fire impacts on hydrological dynamics, we assume that precipitation within a month occurs at one event and the soil water that exceeds the soil water capacity is treated as runoff. This might overestimate the runoff since it is highly possible that there is more than one precipitation event within a month. Second, the calculation of evapotranspiration is based on the Penman-Monteith equation, but with some simplifications. Wind speed, used in the equation, can influence the speed of evapotranspiration while TEM does not consider that. In addition, TEM only considers the evapotranspiration in upper layers of the soil while deep layer soils might also impact this process if roots can reach those layers. Third, TEM runs independently for each grid cell. There are no exchanges between adjacent cells while water runoff is highly affected by the lateral water exchanges. Fourth, fire can affect post-fire runoff through changing soil structure and texture (Shakesby & Doerr, 2006; Ebel & Martin., 2012), but this has not been considered in the current model.

**5 Conclusions**

Driven with satellite-derived burn severity data, our model simulations show that increasing burn severity warms regional mean soil surface temperature by 0.1-0.5 $^{\circ}$C during 2003-2016 in Eurasia northern high latitude forests. Fires decrease post-fire evapotranspiration and increase soil water content, leading to a higher runoff of 131 million $m^3$ $yr^{-1}$ in this region. Wildfires also significantly remove ecosystem carbon through combustion emissions of 121.9 Tg C $yr^{-1}$ and reduce post-fire ecosystem production from 106.4 to 66.1 Tg C $yr^{-1}$. As a result, wildfires cause those forests to lose 2.3 Pg C during the study period, resulting in a shift from a carbon sink to a source. This study highlights the importance of burn severity in modeling soil thermal, hydrological regime and regional carbon dynamics in northern Eurasian forests.

**Conflict of interest**
The contact author has declared that none of the authors has any competing interests

**Acknowledgement**
This study was supported by a NASA grant #80NSSC21K1710.

**Open Research**
The data that support the findings of this study are openly available at the following:
Zhuang, Q. and Xu, Y.: Quantifying the impacts of wildfires on soil thermal, hydrological and carbon dynamics in northern Eurasia from 2003 to 2016, doi: 10.4231/ZJM7-A207,2024



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
