# Peer review of "Quantifying the impacts of wildfires on soil thermal, hydrological and carbon dynamics in northern Eurasia from 2003 to 2016"

_EGUsphere, 2024_

## Author Comment (AC1)

Reviewer 1:

How much of a reduction in litter post-fire is there? Fire burns some portion of litter, of course, and the reduction in LAI means there's less litter being generated. But is any of that lost LAI dropped to litter? It might be helpful to see a time series of net litter flux in burned areas before and after fire.

*Response: Thanks for the comments. In our model, litters are added to soils as soil carbon. Fire will directly remove portion of vegetation and soil carbon based on the burn severity. We assume there is no extra litters from vegetation after fires into soils. As a result, soil organic C decreases after fires depending on burn severity due to burn (Figure 1).*

[Figure]

*Figure 1 Changes of soil carbon before and after fire under different burn severities (dNBR values).*

Does litter have an insulating effect like moss does?

*Response: Yes. Litters are treated as upper organic soil layer, which have different thermal properties from mineral soils to influence thermal dynamics (See Zhuang et al., 2002).*

**Zhuang, Q.**, A. D. McGuire, K. P. O'Neill, J. W. Harden, V. E. Romanovsky, J. Yarie (2002) Modeling the soil thermal and carbon dynamics of a fire chronosequence in Interior Alaska, *J. Geophy. Res.*, 107, 8147, doi:10.1029/2001JD001244.

I was surprised to see negative dNBR values. I see now that's mathematically possible, but how should a reader interpret such values?

*Response: A reader should interpret such values as enhanced regrowth. Negative values are found in areas where the vegetation in the post-fire image has greened up*

*significantly after the fire, thereby making a strong signal in the NIR part of the equation. It can also occur if there was no available post-fire image for quite some time after the burn. It is not uncommon to see these values in areas that have graminoid cover that regenerate relatively quickly as long as the root mass was not disturbed. It is a place that should be put into the "burned" category - low severity.*

Any ideas why fire might decrease soil temperature in some places?

*Response: Fire removes moss layer and litters that serve as heat insulation, leading to higher soil temperature in summer.  Fires also influence the soil hydrological process and surface energy exchange, such as sensible heat flux, thus soil surface temperature that is used to drive soil thermal model, especially in nongrowing season, the less insulative materials due to fire will cool the soil surface temperature, leading to cooler soil temperature, leading to cooler annual soil temperature in some places (Xu and Zhuang, 2023).*

**Xu, Y.** and **Q. Zhuang** (2023), The importance of interactions between snow, permafrost and vegetation dynamics in affecting terrestrial carbon balance in circumpolar regions, Environ. Res. Lett. 18 044007, DOI 10.1088/1748-9326/acc1f7

Minor comments/corrections

P3L19: Citation needed for Landsat data?

*Response: Thanks for the advice. We have added the link.*

P7L1-2 (Fig. 2 caption): Is gray nonforest/not simulated?

*Response: Grey areas are not simulated since there are no fire records in these places.*

P11L21: The "previous estimates" referred to are the Pan-Arctic ones, yes? This could be mentioned in this sentence for clarity.

*Response: The 'previous estimates' refers to our study on fire combustion emission in North American boreal forests (Xu et al. 2024). Combined with this study, we can have*

***an estimation of the whole pan-arctic region in the future. We have revised this
sentence.***

Xu, Y., Zhuang, Q., Zhao, B., Billmire, M., Cook, C., Graham, J., ... & Prinn, R. (2024). Impacts of wildfires on boreal forest ecosystem carbon dynamics from 1986 to 2020. *Environmental Research Letters*, *19*(6), 064023.

P12L10: I think this should be equation 4, not 2.

***Response: Thanks. We have corrected this.***

P13L8-9: Worth pointing out that this limitation doesn't apply to your per-watershed analysis (Table 2).

***Response: When calculating the runoff differences in each watershed, we simply
calculate the sum of runoff differences in each grid cell that is close to the river. The
exchange process might have some non-linear impacts on runoff estimation. Thus,
we believe this limitation might still affect the analysis in each watershed.***

---

## Author Comment (AC2)

Reviewer 2:

The authors offer absolutely no evaluation of model performance for the study region. As the authors state, the parameterisation was developed for North America not Eurasia. The two regions have different ecologies – including different fire regimes and different species with different fire response strategies (i.e. embracer vs resistor strategies) (Rogers *et al.* 2015). The quoted evaluation for North America was fairly light – three sites with large error bars. The original site level evaluation of Zhuang et al. (2002) is good but only goes so far because 1. it only covers black spruce sites (a species not found in Eurasia) and 2. doesn't include the 20+ years of satellite data now available. Firstly, are there site measurement from Eurasia that could be used? But secondly, failing that, there are now a plethora of products that can be used to assess the model performance over the current study region. MODIS GPP/NPP and GOSIF GPP would be amongst the obvious choices, but also (depending on exactly what is prescribed as input to the model and what is calculated) LAI, treecover, biomass and FAPAR could be used. Also soil moisture products as the authors quote soil moisture results, and GLEAM could be used for hydrological variables (although it is a modelled product that might better be used for context, see point 2. below). There might be some reasons why the benchmarking I suggest is inappropriate, or perhaps the authors can explain why the parameterisation for the North America is suitable for Eurasia, but they must address this.

*Response:  Thanks for the comments.  In this revision, we evaluated our simulation results with site observations at three sites in Eurasia, including GPP (for carbon) and soil moisture (for hydrological regime) (Fig. 2).   We also added both satellite-based GPP and soil moisture comparison at regional scales as evaluation of our model results. These comparisons indicate the model and its parameterizations generally capture the observations of carbon and water fluxes.  We added these comparisons into manuscript.*

[Figure]

*Figure 2 Comparison between simulated (blue lines) and observed (red lines) GPP (a, b, c) and soil moisture (d, e) at three different sites (There is no soil moisture observation at 62.5N, 129E).*

*We also compared our simulated GPP with GOSIF GPP in Fig. 3 and simulated soil surface moisture with GLEAM in Fig. 4. We can see that the simulated GPP generally matches GOSIF data in Asia while there is a difference in Europe. The difference might be due to that we simply assume the plant function type as boreal forest which is not consistent with the main plant function type there. However, most of our fire records are in Asia, we believe that this will not have a big impact on our regional estimation of carbon emissions. For soil surface moisture, TEM generally agrees with GLEAM and the differences between them are relatively small for most grid cells.*

[Figure]

***Figure 3 Comparison between TEM simulated and GOSIF mean annual GPP during 2003-2016***

[Figure]

***Figure 4 Comparison between TEM simulated and GLEAM mean soil surface moisture during 2003-2016***

The results badly lack context in terms of numerical magnitudes. Some of this could be address by including observed values (see above) to give context, but in many cases just a little bit of extra information (such as baseline values without fire) is needed to make the tables and figures actually informative. For example:

Figure 5. Each of these maps should be presented as both the absolute difference and relative change. For panels a) and b) it is important to know if these changes actually amount to much in relative terms (I feel that the evapotranspiration changes are likely negligible, even though the result is quoted in the abstract). For panel c), the LAI changes look considerable, but it would be important to know what the baseline was.

***Response: We added the relative change in the Figure. We also fixed a bug in our code and recalculated the variables. Now, we can see that all the three variables in the Figure exhibit significant changes under fire's influence.***

Table 2 – these numbers differ by three orders of magnitude! Of course different basins have different sizes so this is understandable, but please give the simulated baseline (and ideally gauging station data if available) so that we can assess these changes in context.

***Response: We added the relative changes of runoff in the table.***

Table 3 – there is no reason that the authors can't subset their estimates to the regions of "Russia", "Eurasia", "Pan-arctic" and "Siberia" to better put their results in context of the others. There are also global fire emissions products – GFAS, GFED etc – that can be easily calculated for the region and included here.

***Response: We reorganized the table and added the results from GFED4.***

The abstract result of a reduction of 131 million $m^3$ $yr^{-1}$, how much is that as a proportion of the current run-off?

***Response: The new estimated regional runoff difference is 37.0 $km^3$/year, accounting for about 41.8% of the runoff when there is no fire. We added the analysis in the manuscript.***

Much methodological detail is missing. Details of how the plant functional types were derived and prescribed are needed. Also details of how fire was prescribed (presumably applied to a fraction of a gridcell only?) are missing. How are photosynthesis and fapar calculated? And respiration? How is vegetation represented - average individual or cohorts or something more abstract? The description of the processing of the GFA data is admirably complete (full marks there), but at least a passing description of the other aspects of the model are required.

*Response: For simplicity, we assume the plant functional type as boreal forest for the whole study area. Fire's influence is calculated based on the area-weight mean of each fire event. TEM does not model dynamic vegetation. In TEM, vegetation carbon is considered as a pool and TEM does not consider the carbon allocation in the pool. Details of how to calculate the photosynthesis and respiration are added in the Method section.*

It would is also important to know how the parameters (of which there are quite number, at least p1, p2, a1, b1 and gamma) were derived and what is there values. There is a small attempt to examine parameter uncertainty by changing values by 10% but we don't have any idea what the uncertainty on the paramaters actually is and only p1, p2, p3, and p4 are tested. What about a1, b1 and gamma? And possible other parameters that are not introduced in the current text? And the p3 is quoted as being the major source of uncertainty – but what about the amount of biomass that is present to burn? That will be a huge factor, but we have no idea how that was calculated (new mind how much below ground and how much above ground) and what the uncertainty on that is. Without further information, I would guess this is the largest source of uncertainty in the study.

*Response: We added descriptions and values of the parameters. We use dNBR values to estimate CBI and burned proportion of vegetation and soil carbon, which is the core of our study. That's why we want to test the sensitivities of the four parameters. Among the four parameters, P3 might have the largest uncertainty. The amount of biomass might lead to larger uncertainties. We revised the description in the Discussion from 'P3 might have the largest uncertainty in this study' to 'P3 might have the largest uncertainty in this method'. Also, we mentioned that biomass amount estimation might be a large uncertainty in the discussion.*

At the end of the day, the lack of evaluation is the biggest problem here. Without it, the results given here are just model output about which it is hard to have any confidence.

There are a plethora of datasets out there that can be used to evaluate the model or at least give context. The lack of methodological details and context also inhibit confidence. These three points also combine to give a lack of clarity on what is the real novelty of the results. Global models (fire-enabled DGVMs for example) could also be used to produce these results. These models would likely not show particularly good model skill, so using a regional model is definitely a good idea. But without model evaluation and greater methodological clarity, one can't be sure that the application of TEM (uncalibrated for the study region) is really an improvement. On account of the these three points, I am suggesting major revisions. If these points are comprehensively addressed I am willing to review the manuscript again as I believe it shows definite potential.

*Response: In all, we added model validation by comparing with site observations. We have also evaluated our model results with satellite data as much as we can. We also gave relative changes of the variables and revised the manuscript with more methodological details.*